# 3D Printing and Its Current Status of Application in Obstetrics and Gynecological Diseases

**DOI:** 10.3390/bioengineering10030299

**Published:** 2023-02-27

**Authors:** Caihong Hu, Weishe Zhang, Ping Li

**Affiliations:** 1Department of Obstetrics, Xiangya Hospital, Central South University, Changsha 410008, China; 2Hunan Engineering Research Center of Early Life Development and Disease Prevention, Changsha 410008, China

**Keywords:** bioprinting, gynecology, obstetrics, 3D printing

## Abstract

3D printing technology is a novel method of utilizing computer-generated three-dimensional models for drawing, assembling special bioinks, and manufacturing artificial organs and biomedical products. In recent years, it has evolved into a relatively mature therapeutic approach and has been widely used in clinical and basic research. In the field of obstetrics and gynecology, 3D printing technology has been applied for various purposes, including disease diagnosis, treatment, pathogenesis research, and medical education. Notably, researchers have gained significant application experience in common gynecological and obstetrical disorders, such as intrauterine adhesions, uterine tumors, congenital malformations, and fetal congenital abnormalities. This review aims to provide a systematical summary of current research on the application of 3D bioprinting technology in the field of obstetrics and gynecology.

## 1. Introduction

Three-dimensional (3D) printing is an extended application of rapid prototyping or additive manufacturing. It is a technology that is based on the principle of layered manufacturing for the processing of materials [1] and precise-control molding of materials in a layer-by-layer manner [2]. This interdisciplinary science is closely related to the fields of medicine, biology, mechanical engineering, and materials science. Broadly, the process of 3D printing occurs in four stages, which can be distinguished based on the types of biomaterials used at each stage, as shown in Figure 1 [2]. The 3D bioprinting process involves the simultaneous printing of living cells and biological materials, using a computer-aided transmission process to print the bioengineered structure [3,4], and the manipulation of cell-loaded bioink to create living structures [2]. Inkjet printing was the first bioprinting technology to be developed [5]. Bioprinting has been used to produce organs—such as the pelvis, mandible, and other supporting structures—in applications related to tissue regeneration technology and surgical simulation [6]. Currently, the 3D printing technology developed in the 1980s is commonly used for bioprinting, and functions on the basis of specific digital designs created through data derived from conventional two-dimensional computed tomography (CT), ultrasound (US), and magnetic resonance imaging (MRI) images. This process involves several steps required for image processing and format conversion, to obtain solid 3D models. While the earliest applications of 3D printing were in industrial fields, after continuous development and progress it has gradually been applied to medicine-related fields [7].

### 1.1. Types and Characteristics of 3D Bioprinting Techniques

The process of 3D bioprinting includes three steps: (1) the formation of target images [8], (2) tissue printing, and (3) tissue structure culture [9]. Common printing methods include inkjet bioprinting [10], extrusion bioprinting [11], laser-assisted bioprinting [12], acoustic bioprinting, lithographic bioprinting, and magnetic bioprinting [13,14]. These bioprinting methods can be used individually or in combination to achieve the desired manufacturing goals of associated organizations (Table 1). Nevertheless, the resolution of the printed structures is affected by the printing material, the operating temperature, and the size of the nozzle.

### 1.2. Selection of Materials for 3D Bioprinting

The construction of in vitro models requires three components that are considered as the trinity necessary for tissue engineering, namely biomaterials, cell types, and stimuli. To create these constructions, biomaterials such as bioinks and related polymers are needed. Biomaterials composed of cells, basic structural materials, and other essential components can form bioinks [15]. They can be in solution or hydrogel form and are loaded with the desired type of biological cellular material considered to be necessary for bioprinting and the development of functional tissue or organ structures [16,17]. Their roles can be classified into four categories: structural bioinks, free bioinks, scaffold bioinks, and functional bioinks [18].

A range of biomaterials is currently available—including natural biomaterials, synthetic biomaterials, and complex biomaterials—which can increase cell proliferation, motility, and differentiation levels. Natural biomaterials provide biological cues similar to those observed in vivo, but their precise control is challenging. In contrast, synthetic polymers can be controlled more effectively during experiments but require modification to provide biological cues.

The most commonly used natural biological materials include fibrin and matrix material (Table 2) [19,20]. Collagen is the most abundant component of the extracellular matrix (ECM). Type I collagen is the main structural protein, and type IV collagen is a key component in cells where differences in cell shape, structure, and function can be observed. The matrix facilitates many types of cell attachment and proliferation processes: it provides information to the cell regarding the ECM and growth factors, stimulates cell-matrix interactions, and induces differentiation. However, while synthetic biomaterials are biocompatible, biodegradable, and renewable, owing to their bioinert nature, unmodified synthetic biomaterials often fail to support desired cell behavior and tissue formation (Table 3) [19].

### 1.3. Application of 3D Printing Technology in Clinical Medicine

Improvements in 3D printing technology and materials used for printing have culminated in a remarkable increase in the expansion of use of 3D printing technology in medicine. In fields such as orthopedics [21], cardiovascular medicine [22], neurosurgery [23], and head and neck external otolaryngology [24], personalized precision medicine is used during teaching, preoperative planning, surgical simulation, and repair and reconstruction. In recent years, 3D printing technology has gradually been utilized in obstetrics and gynecology.

## 2. Applications of 3D Printing Technology in Obstetrics and Gynecology

Obstetrics and gynecology are two medical disciplines involving the study of physiological and pathological changes in the female reproductive system. Diseases affecting the female reproductive system that occur most commonly include diseases of the internal genitalia, including neoplastic diseases (uterine fibroids, cervical cancer, endometrial cancer, ovarian cancer, etc.), endocrine diseases (premature ovarian failure, polycystic ovary syndrome, abnormal uterine bleeding, etc.), structural diseases (intrauterine adhesions, congenital genital tract malformations, etc.), pelvic floor support dysfunction-related diseases (pelvic floor organ prolapse, etc.), and perinatal diseases (congenital fetal malformations, placental abnormalities, etc.). In recent years, use of 3D printing technology has gradually developed as a new therapeutic approach and strategy for the treatment of obstetrics and gynecological diseases (Table 4).

### 2.1. Tumors

#### 2.1.1. Uterine Fibroids

Uterine fibroids are the most common benign tumors of the female genitalia, particularly among women of reproductive age. The growth of fibroids may result in compression-related symptoms, secondary anemia, miscarriage, and infertility. Currently, the primary treatment available for uterine fibroids mainly involves surgical resection.

3D printing technology can help to guide the clinical diagnosis and treatment of uterine fibroids in pregnant and non-pregnant patients. In combination with preoperative US, CT, MRI, and other related examinations, 3D printing technology has been applied for preoperative simulation of uterine fibroids [25]. This technology can assist surgeons in determining the surgical plan, incision size, and other presurgical treatment. For example, a patient with ossified fibrosis successfully underwent uterine and bilateral adnexectomy with the assistance of preoperative 3D printed models and recovered effectively after surgery [26]. Uterine fibroids are the most common form of benign tumors during the reproductive period, and may cause increased local blood supply, intrapartum atony, and postpartum hemorrhage during pregnancy. Three-dimensional printing technology can also be used to determine the optimal site of an abdominal and internal incision [27] before surgery, helping clinicians to develop a plan in advance and reduce the risk of intraoperative bleeding. Mackey et al. [14] successfully developed a 3D printed model using MRI data from a pregnant woman with multiple uterine fibroids as an evaluation tool for a planned cesarean section. This model accurately depicted the number, size, and location of uterine fibroids and helped to determine the optimal location for skin and uterine incisions. Finally, the patient underwent cesarean section at 37 weeks and 7 days of pregnancy with a J-shaped incision on the uterus, by comparing the actual anatomy with its 3D printed mode. Ultimately the intraoperative bleeding was limited to approximately 600 mL and there were no related complications, resulting in good maternal and infant outcomes. Moreover, the use of a 3D workspace can facilitate the visualization of the uterus, placenta, and fibroids in different colors, providing guidance for doctors in administering surgical treatment.

#### 2.1.2. Malignancies

Malignant tumors of the female reproductive system are an important cause of female mortality, with the mechanism associated with their occurrence being still unclear. Surgical resection, radiotherapy, and chemotherapy are the three most effective methods used for the treatment of malignant tumors; however, patients outcomes exhibit differences. Hence, personalized treatments—including the formulation of surgical plans, location and scope of radiotherapy, and choice of chemical—are required. The development of 3D printing technology provide a potential solution to the problem of sample shortage, facilitating simulation of the tumor microenvironment. This is conducive to the study of pathogenesis and targeted drug selection.

(1)Cervical cancer

Cervical cancer is the most common gynecological malignancy, and its onset has been reported to occur at a younger age in recent years. A 3D printed cervical model prepared by combining the patient’s imaging data, can provide not only clear assessment of the depth of tumor invasion and surrounding lymph node metastasis, but also potential treatment strategies for patients with advanced cervical cancer. Research shows that a 3D printed model composed of cervical cancer cell lines and hydrogels composed of gelatin, alginate, and fibrinogen can simulate the tumor microenvironment, facilitating the study of tumor pathogenesis and metastasis [28]. Three-dimensional printing models can assist clinical chemoradiotherapy strategies and formulate individualized treatment plans. Digital modeling and 3D printing technology can be used for pre-planning radiotherapy to optimize drug administration during radiotherapy. This can ensure the efficacy of radiotherapy and improve patient outcomes [28]. Li et al. [29] successfully personalized 3D molds of 30 patients to assist with treatment of cervical cancer, demonstrating that this technology could help to improve local tumor control rates and reduce exposure of healthy organs to radiation. Moreover, the further reconstruction of MRI data from the pelvic autonomic nerve and related organs in cervical cancer patients in 3D could significantly improve nerve-sparing surgery by characterizing the extent of nerve distribution and the anatomical relationship of the uterus with the surrounding organs.

(2)Endometrial cancer

Endometrial cancer is a malignant epithelial tumor that commonly occurs in the endometrial gland, making it one of the three major malignancies of the female reproductive tract. It is caused by a hormonal imbalance, and its incidence has been increasing worldwide in recent years. In vivo and in vitro models help in understanding the mechanism of tumor invasion, thereby providing new strategies for the clinical treatment of endometrial cancer. In vitro models can help in understanding the invasion and progression of these tumors and in developing more effective treatment strategies in the future [19].

At present, comprehensive staging surgery is the most commonly used surgical treatment for endometrial cancer, and the ureter is the most vulnerable anatomical structure during surgery. The application of 3D printed models can be simulated before surgery to avoid damage during surgery. However, there may be some differences between the 3D printed model and the actual situation. Sayed Aluwee et al. [30] analyzed the anatomical structure and uterine lesions of patients based on their preoperative MRI images, and then combined 3D printing technology with model casting technology to create an accurate hysterectomy model for endometrial cancer patients that could effectively demonstrate the complex 3D structure of the human uterus with an error of 1.19 to 2.22 mm. This model could improve the effectiveness and accuracy of preoperative planning and help patients to better understand their disease, the surgical process, and the risk of complications. Postoperative recurrence of endometrial cancer occurs most commonly in the vaginal area, and personalized 3D molds could help to provide personalized treatment to individuals undergoing postoperative chemotherapy while minimizing damage to surrounding organs, as seen in the case of cervical cancer.

(3)Ovarian cancer

Several different tissue types of ovarian cancer, including high-grade serous ovarian cancer, clear-cell carcinoma of the ovaries (OCCC), endometrioid ovarian cancer, mucinous ovarian cancer, low-grade serous ovarian cancer, and others, are characterized by specific molecular events and clinical behaviors.

In addition to aiding preoperative simulation and intraoperative guidance, 3D printing technology can be utilized in basic research to facilitate the assessment of chemotherapy drug sensitivity and optimal drug concentration, thereby enabling the formulation of individualized treatment plans and minimizing adverse drug side effects. It is possible to use 3D-printed drug models, personalized treatment, and on-demand drug delivery to improve the safety and effectiveness of medications for patients [31,32]. Cisplatin (CDDP) is a commonly used chemotherapy drug, but its effectiveness varies depending on the tissue type. Therefore, it is not necessary to find an appropriate individualized reference dose; however, CDDP does have certain side effects and is associated with drug resistance in clinical applications. Wang et al. [33] used 3D printing technology to develop a drug-loaded stent by encapsulating CDDP in a PLGA stent, which was directly implanted near the tumor site. The drug-loaded stent had a high local drug-release capacity, which improved bioavailability, prolonged controlled drug release, and reduced side effects.

In vitro and in vivo experiments have demonstrated that drug-loaded stents could achieve tumor suppression more effectively at relatively low doses, compared with intraperitoneal drug injections. Thus, they could be used as a new mode of tumor treatment. In addition, the biomimetic in vitro 3D model used for co-culture more accurately represents in vivo phenomena, allowing for the more precise prediction of patient cellular responses to chemotherapy drugs. The culturing of cells derived from the patient required only a short amount of time and had a high success rate. Moreover, the accuracy of predicting cell response to chemotherapy drugs using this model was 89% [34].

Notwithstanding the above, the existing 3D bioprinting model still lacks common tissue structure components, such as the vascular system and immune system, that are found in vivo. Accordingly, future studies should consider incorporating relevant components to summarize the microenvironment more effectively, providing more reference information, and developing precision medicines.

### 2.2. Premature Ovarian Failure

Premature ovarian failure (POF) is characterized by early deterioration of ovarian function, resulting in reduced oocyte quantity and quality and decreased hormone levels, ultimately leading to infertility and difficulty conceiving, as well as various comorbidities, such as obesity, diabetes, Alzheimer’s disease, genitourinary atrophy, osteoporosis, fractures, and cardiovascular diseases. Hormone replacement therapy (HRT) is a current mainstream treatment used to relieve symptoms and avoid long-term complications, but it is associated with an increased risk of thrombosis, cancer, and stroke. The development of bioengineering techniques has facilitated the gradual implementation of 3D printing for studies of treatments for POF. Biomaterials currently available for the study of POF include exosomes, extracellular matrix, collagen, etc. Among these, collagen is a promising hydrogel that can wrap itself around ovarian follicles. The use of different kinds of biological materials should ensure cell growth and improve survival time. Sharma et al. [35] first developed a 3D collagen gel culture system to study the in vitro growth and survival of animal prefollicles. It was found that a medium supplemented with growth factors gave a higher survival rate and larger follicle diameter on the tenth day of culture. Furthermore, the cultured runaway could survive for up to 20 days, a marked improvement compared with a survival time of 10–15 days in the control group.

Cell growth in 3D bioprinting is influenced by the concentration of biological material. Joo et al. [36] created a bionic 3D shell, rich in collagen, for culturing rodent ovarian follicles in various concentrations of type I collagen ranging from 1% to 7%. The study showed that in a relatively high concentration of collagen, a denser matrix is produced. In collagen hydrogels of 3 mg/mL and 5 mg/mL, the survival rate of ovarian follicles was over 90% (90 ± 4% and 98 ± 2%, respectively). Alterations in collagen hydrogel density and elasticity resulted in variations in follicle growth and development, sex hormone production, and oocyte maturation, emphasizing the importance of physical properties in the maintenance of the phenotype and function of follicles in 3D culture systems [37].

Wu et al. [38] used gelatin-methacrylate ester (GelMA), alginate, and gel-alginate as bioinks to fabricate 3D printed scaffolds for ovarian cells. They used ovarian tumor cell lines (COV434, KGN, ID8) and primary cultured ovarian somatic cells for 3D printing and cell-loaded bioprinting, respectively. They proved that GelMA is more suitable for the 3D printing and processing of ovarian cells through material expansion, enzyme degradation, and in vivo culture. GelMA scaffolds performed effectively in terms of swelling, degradation power, and fidelity. Moreover, there was almost no primary cell death in two-dimensional cultures prepared using GelMA, oocytes were observed at all developmental stages in experiments conducted in vivo, and the follicular cell survival rate of in vitro culture after 7 days was 84%.

Nevertheless, extrusion-based cell-loaded bioprinting resulted in a cell death rate of 90%, suggesting that extrusion 3D culture was not suitable for culturing primary ovarian cells. Thus, 3D-printed scaffolds were found to create a suitable microenvironment for follicle development, thereby providing a new strategy for in vitro follicle growth.

### 2.3. Intrauterine Adhesions

Intrauterine adhesions (IUA), also known as Asherman’s syndrome, are caused by. damage to the basal layer of the endometrium, endometrial adhesions, or fibrosis, which result in the partial or complete atresia of the uterine cavity [39]. It mainly manifests as intermittent abdominal pain, decreased menstrual flow, and even amenorrhea [40], and is a significant cause of infertility in women of childbearing age. The formation of intrauterine adhesions is directly related to damage of the basal layer, which is where the endometrial stem cells are located. Hysteroscopic intrauterine adhesiosiotomy (TCRA) is the main method for treatment of IUA, but re-adhesion may occur after the procedure. In recent years, with the continuous development of regenerative medicine, the transplantation of exogenous stem cells has represented a new treatment option for IUA [41].

With regard to IUA treatment, current studies have proven that granulate colony stimulating factor (G-CSF) has a positive effect on endometrial regeneration, despite the fact that G-CSF has a short half-life in the human body. Because of internal migration, its concentration at the injury site is not high. Therefore, sustained-release drug delivery systems with high local drug concentrations should be investigated. Wen et al. [42] prepared a 3D-printed scaffold with gelatin and sodium alginate as raw materials, along with a G-CSF-SRM (3D microsphere) sustained-release system with an average diameter of 9.68 μm. In vitro experiments have shown, using a sustained-release system in the IUA rat model, that such an approach could reduce adhesion significantly after an endometrial injury, promote the reconstruction of the endometrial structure and function, and aid in achieving spatial control and personalization of drug distribution. In addition, the 3D microspheres in the Sprague–Dawley IUA rat model could also be used to promote the local regeneration of the endometrium, improve its receptivity, and preserve the pregnancy-related functions of the damaged endometrium. In the rat model of intrauterine adhesion described by Salama et al. [43] the experimental group was composed of IUA model rats treated with bone marrow mesenchymal stem cells, while the control group was composed of IUA rats or untreated IUA rats. Histological staining and immunohistochemical analysis showed significant endometrial thinning, fibrosis, and degeneration of the endometrial epithelium in the control group. In contrast, changes were substantially reversed for the experimental group. These findings suggest that mesenchymal stem cells might be ideal for treating intrauterine adhesions and could provide a novel strategy for future research on IUA treatment using mesenchymal stem cells from various sources.

The purpose of treating IUA is to improve pregnancy rates and relieve the associated symptoms. Ji et al. [44] used a 3D bioprinting device to construct a hydrogel scaffold loaded with human-induced pluripotent stem cell-derived mesenteric stem cells (hiMSCs) and implanted it in vivo. They found that it could promote endometrial tissue recovery, and promote regeneration in endometrial cells, including mesenchymal, epithelial, and nerve cells. The generation process led to improved endometrial function indicators that facilitated embryo implantation and helped maintain pregnancy. The pregnancy rate in the experimental group was as high as 100%, while it was only 15–60% in the control group. These findings suggested that the use of 3D-printed hiMSC-loaded hydrogel scaffolds might be ideal for endometrial repair.

Preventing the recurrence of IUA is the primary goal of post-treatment care. Research has shown that the collagen scaffold (CS) model loaded with human umbilical cord mesenchymal stem cells (UC-MSCs) can promote endometrial structural reconstruction and functional recovery. Local application of CS/UCs after hysterectomy in severe IUA patients prevented re-adhesion, promoted endometrial regeneration, and improved pregnancy outcomes [45]. Feng et al. [46] used 3D bioprinting technology to construct composite tissue hydrogels with different gelatin-methacrylate (GelMA) and collagen methacrylate (ColMA) ratios. In vitro experiments revealed that in 3D printing processes performed using this hydrogel, stem cells could be released continuously in vitro for at least 7 days. In vivo experiments demonstrated that a hydrogel model composed of GelMA/ColMA/hAMSC (human amniotic mesenchymal stem cells) could prevent intrauterine adhesions in the rat IUA model.

The use of 3D correlation technology to treat IUA could reduce the number of adhesions, improve the shape of the endometrium, increase the blood flow to the endometrium [47], increase the thickness of the endometrium, and increase the pregnancy rate [48]. Three-dimensional printing and tissue engineering may provide new strategies for the treatment of IUA and prevention of their recurrence.

### 2.4. Malformations of the Genital Tract

The treatment of reproductive abnormalities requires definitive diagnosis, instru mental assistance, and adequate preoperative simulation. Three-dimensional printing technology can be used to develop individualized molds for congenital structural abnormalities, which can be used to manufacture devices rapidly, save on the opening costs for mold creation, and solve problems associated with the inability to mass-produce new equipment. For example, vaginal loss may be caused by various congenital or acquired factors and can cause significant physical and psychological pain to patients. The method for its treatment mainly involves vaginal reconstruction technology, as the tissue used in its treatment is of non-vaginal origin, and there may be complications such as contractures, necrosis, prolapse, and malignant transformation after surgery. The application of vaginal tissue can reduce the occurrence of these complications. Hou et al. [49] used decellularized animal vaginal epithelial tissue mixed with gelatin and sodium alginate to produce a bioink with excellent biocompatibility, which was then utilized to print 3D scaffolds for bone marrow mesenchymal stem cell (BMSC) culture. An experiment in vitro found that BMSCs exhibited high survival rates (95%) in 3D scaffolds on the first day of examination and retained high cell viabilities after 7 days. Vaginal epithelial cells were successfully cultured in vivo in mice, indicating that 3D printing technology could represent a new method for the development of vaginal tissue replacements.

However, the reconstruction of the vagina requires not only the incorporation of vaginal epithelial cells but also consideration of the thicknesses and sizes of different parts. In experiments involving mice by Tian et al. [9], the thickness of epithelial cells was measured by HE staining, and that of the smooth muscle layer, propria, and outer membrane was measured via α-actin immunohistochemical staining. The results showed that the lower segment of the vagina was thicker than the upper segment. The difference in thickness and the volume density of the upper and lower vaginal epithelium, lamina propria, and outer membrane, however, was not significant, which provided new insights regarding the 3D printing of vaginal information and vaginal structure. In a case report, pelvic MRI image data and photosensitive resin materials were used to generate a 3D printed model of the reproductive tract of patients, while a preoperative evaluation (including physical examination and 3D printing technology) was performed to confirm the diagnosis of a complete uterine septum with a double cervix and double vagina during hysteroscopic surgery [20]. Thus, the conversion of two-dimensional MRI imaging information into intuitive three-dimensional objects can help clinicians to make preoperative diagnoses and enable patients to understand their abnormalities more effectively. Moreover, it is valuable for providing primary anatomical education about complex genital tract malformations.

### 2.5. Perinatal Medicine

#### 2.5.1. Prenatal Diagnoses of Fetal Malformations

The launch of the three-child policy in China has resulted in an increase in the proportion of elderly mothers and abnormalities such as fetal structural malformations. At present, the evaluation of intrauterine status before delivery is highly dependent on imaging examinations, which are mostly two-dimensional, and their accuracy is affected by a series of factors, such as the fetal position.

Prompt diagnosis of fetal congenital anomalies can guide patients in their decision whether or not to terminate pregnancy in a timely manner, to reduce possible damage to the body and guide subsequent pregnancies. Three-dimensional printing technology can diagnose congenital malformations in the uterus and formulate corresponding treatment plans. 3D printed models of fetal malformations—such as fetal tumors, lymphangiomas, achondroplasia, and congenital cardiovascular malformations—could be used to predict fetal prognosis, determine whether it is necessary to terminate a pregnancy, and inform corresponding genetic counseling services [50,51,52].

Congenital heart defects are the most common congenital malformations. A routine autopsy is the gold standard method used for their diagnosis. However, this method lacks diagnostic ability to a certain extent, especially in individuals with a low gestational age and body weight. Sandrini et al. [13] examined 21 cases of fetal cardiac abnormalities using CT data as input information for 3D printing, prepared 3D printed models for all patients with a ratio of 1:1, and scaled the models by a factor of five. When printed using fivefold magnification, the model could be cut into four chambers to observe and analyze the inner heart structure. Using the 3D printed model, all 21 patients were diagnosed with congenital heart disease and a combination of other malformations.

In a clinical study of 12 fetal abnormalities, Liang et al. [53] successfully printed 3D models for abnormalities of the cleft lip and palate, spine, heart, or brain by including data from a 3D and 4D color Doppler ultrasound. They found that the 3D printing results were consistent with the original images. Although a certain amount of deviation (average deviation of 0.1 mm) was observed, it is widely thought that a model developed using a combination of both 3D printing and a Doppler ultrasound can accurately represent and describe fetal abnormalities, which can significantly reduce the misdiagnosis rate of clinicians (from 5% to 0.4%) and students (from 17.9% to 0.4%, and a reduction to 5.2%). In a case report on fetal facial deformities, 3D printing technology was used to create a fetal facial model, which clearly showed facial deformities, and supported a precise diagnosis of left facial cleft. After a consultation, the parents chose to terminate pregnancy at 24 weeks of gestation [12]. In addition, the combination of a 3D printed fetal heart model produced by fetal echocardiography and MRI blood flow data, can help to visualize and quantify the intracardiac blood flow profile in vitro, which can assist in the diagnosis of congenital cardiac blood flow and structural abnormalities, such as hypoplastic left heart syndrome [19]. In addition, a 3D printed model can be used to study the blood flow pattern in a healthy and abnormal fetal heart and evaluate the impact of morphological changes on the blood flow on the heart [54], to enable the execution of corresponding interventions. In a randomized case-control study, the printing of fetal models together with the explanation of relevant details to pregnant women was proven to reduce anxiety during pregnancy, enhance the maternal–fetal relationship, and improve awareness regarding prenatal checkups [55].

Thus, 3D printing is a highly valuable tool for diagnosing fetal malformations, and can be applied to family consultation, fetal diagnosis, and personalized fetal and neonatal treatment strategies in the future.

#### 2.5.2. Prenatal Assessment of Placenta Accreta Spectrum

Patients with placenta accreta spectrum (PAS) are at risk of severe postpartum hemorrhage, which may lead to hysterectomy, hemorrhagic shock, Sihan syndrome, and even maternal and infant death; thus, they have poor prognoses. The combination of 3D bioprinting and an imaging examination can help to clarify the placental implantation depth, position, and invasion of surrounding tissues, and guide preoperative surgical plans such as uterine incision selection, abdominal wound size, bleeding risk assessment, etc. Additionally, it can be used as a research tool for conservative surgical treatment of patients with placenta implantation spectrum [56].

#### 2.5.3. Clinical Teaching in Obstetrics and Gynecology

Obstetrics and gynecology are unique disciplines with equal emphasis on foundation and practice. The relatively strained relationship between doctors and patients and limited anatomical specimens and clinical practice opportunities limit the rapid growth of young doctors in this field. Three-dimensional printing technology provides novel solutions for addressing this problem. For example, different silicone materials could be used to make a 3D-printed model of a postpartum perineal laceration that could be used to provide training on suturing to physicians [57]. The use of 3D-printed embryo models with different gestational ages and proportions can help to understand the dynamics of embryonic development, which lays a foundation for deriving more knowledge in this field [58,59,60]. Moreover, the use of computer-aided design software to create cervical prototypes and add tumor masses and vascular openings could help to realistically simulate a model of cervical cancer that exhibits bleeding [16]. These 3D printed models have been used to teach trainees effectively.

**Table 4 bioengineering-10-00299-t004:** The applications of 3D printing technology in obstetrics and gynecology.

Classification	Disease	Application	Models	Strength
Benign tumor	Uterine fibroids [14,26,27]	Preoperative simulation, intraoperative guidance, teaching	3D-printed tumor models	Relationship between fibroid sites and surroundings can be clarified Different colored markers can be applied to distinguish different tissues
Malignancy	Cervical cancer [28,29]	Preoperative simulation, intraoperative guidance, teaching, radiotherapy and chemotherapy guidance, basic research	3D-printed tumor and surrounding tissue models, in vitro cultured cell models, in vivo cultured animal models	Identification of the invasion of the lesion and surroundings Guidance for surgical planning to minimize physical injury Development of individualized chemotherapy regimens Provision of more models for basic research
	Endometrial cancer [30]			
	Ovarian cancer [33,34]			
Functional diseases	POF [35,36,38]	Treatment	3D printing of ovarian tissue model	Radical treatment
Structural diseases	IUA [42,43,44,45,46]	Treatment, prevention	G-CSF-SRM hiMSC-loaded hydrogel scaffolds	Stable release Long duration of action
	Malformations of the genital tract [9,49]	Diagnosis, teaching, individualized molds, preoperative simulation, intraoperative guidance	3D printing of abnormal organ models	Individualized treatment plan
	Fetal malformations [52,53,54]	Diagnosis, treatment, teaching	3D printed fetal models	Can be magnified at different scales
	PAS [55]	Diagnosis, preoperative simulation, intraoperative guidance	3D printed tissue models of the uterus and placenta	Determine the extent of placental implantation, surrounding tissues and angry blood vessels to reduce intraoperative bleeding

## 3. Discussion

3D printing technology is a powerful tool that allows for the transformation of a vir tual three-dimensional image to a physical, solid model using a combination of visual and tactile feedback. This technology has been widely used in several departments such as orthopedics, cardiovascular surgery, obstetrics and gynecology, and plastic surgery. In addition, the application of the 3D printing model for teaching purposes could help students achieve a better understanding of human anatomy through varied perspectives. The dual combination of vision and touch could help students understand the general characteristics of the human body structure more effectively, thereby improving teaching quality and their enthusiasm for learning and improving their ability to link theory to practice and think critically. Notably, the use of 3D printing technology has also been highly advantageous as it has provided ideas for addressing the problem of organ shortage. The study of the pathogenesis of different diseases requires cells to be derived from the body, but it is not easy to obtain the required number of human cells for such analyses. We can obtain sufficient amounts of tissues using 3D printing technology for studies on tumor pathogenesis and the sensitivity of different types of tumors to chemotherapeutic drugs, which can provide guidance regarding clinical diagnosis and treatment.

The continuous development of 3D technology has played a substantial role in obstetrics and gynecology, with applications in preoperative simulation; adjuvant radiotherapy and chemotherapy; and the diagnosis of congenital genital tract malformations, fetal malformations, and placental implantation spectrum diseases. Because of advantages such as high reproducibility, safety, and repeatability, 3D printing technology has been applied increasingly in various disciplines at home and abroad. However, the technology is time-consuming, and there are limitations associated with its relatively high cost, lack of trained professionals, material selection, biomechanical control, and long-term survival of printed constructs, which limit its application in clinical practice.

3D printing technology has helped us achieve substantial results in obstetrics and gynecology, but its applications have been limited. For example, placental and vascular causes might result in complications such as twin-to-twin transfusion syndrome and selective fetal growth restriction in complex twin pregnancies. The diagnosis of these disorders is currently dependent primarily on ultrasound imaging techniques. Treatment necessitates the blocking of the vascular connection between twins, and fetoscopic surgery is the most common method used in such cases. Three-dimensional printing technology could thus be applied for the development of a preoperative surgical plan and simulation, and for monitoring of intraoperative adjuvant and postoperative treatment effects. In order to select the optimal delivery mode, the fetal model and the maternal pelvic model could be printed at equal scales, and magnetic resonance and other imaging data could be obtained. These may be used in combination with the results of a head and pelvic evaluation, to determine the clinical delivery mode. Notably, 3D printing technology has not been fully utilized in research on tumor pathogenesis, tumor metastasis, tumor treatment, etc., and biomaterials exhibiting a better performance and stronger biomimetic ability should be fully explored for 3D printing in the future.

3D printing technology can provide insights regarding a new direction for accurate disease diagnosis (Figure 2) and personalized treatment (Figure 3), which can not only facilitate the study of the mechanism of disease occurrence and development and simulate the activity of drugs in the human body, but also reduce the number of laboratory animals involved in experiments. However, progress in 3D bioprinting technology in obstetrics and gynecology is still in its infancy due to certain complications (biological toxicity), high printing time, increased economic costs, and limited accuracy of printing materials; hence, these aspects need to be explored further. In the future, we should continue to rapidly explore relevant materials; formulate industry standards; standardize medical equipment manufacturing; and diagnose auxiliary diseases, preoperative designs, and clinical experimental applications. The application of printed cells and tissues to the tissue engineering process would enable us to gradually develop a mature new technology for the integration of appropriate biomaterials, cell types, and stimuli, i.e., the new trinity of factors necessary for performing tissue engineering using 3D biomaterial technology.

## Figures and Tables

**Figure 1 bioengineering-10-00299-f001:**
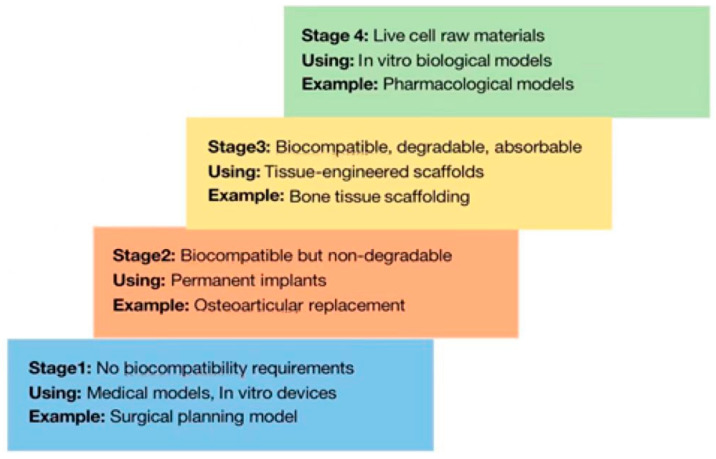
Different stages of development of 3D printing [2].

**Figure 2 bioengineering-10-00299-f002:**
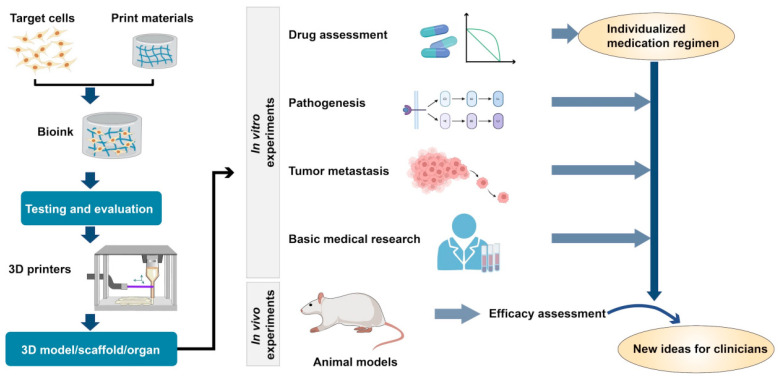
The process of 3D printing and application of tissue and organ models.

**Figure 3 bioengineering-10-00299-f003:**
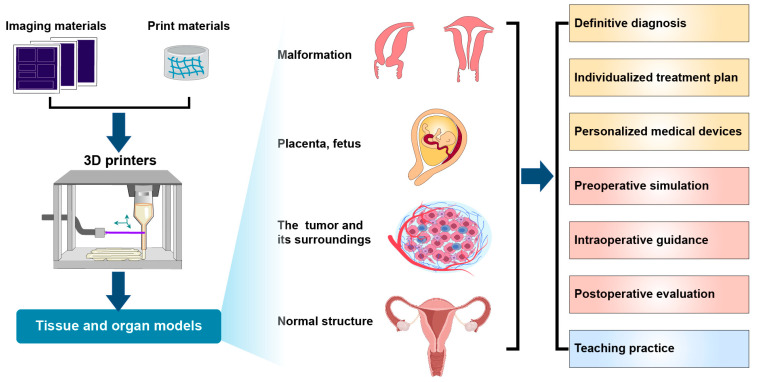
The process of 3D printing and its applications.

**Table 1 bioengineering-10-00299-t001:** Differences in various types of bioprinting processes [10,11,12].

Characteristic	Inkjet Bioprinting	Extrusion Bioprinting	Laser-Assisted Bioprinting
Principle	Droplet form of thermal or sonic methods	Linearly applied pneumatic pressure or mechanical force	Laser-induced forward transfer
Strengths	Fast speed, low cost, wide availability, and high cell viability	Deposition of high-density cells with high structural integrity	Enables the printing of different living cells and biological materials with precision and micron-level resolution
Limitations	Low pressure and easily blocked nozzle	Pressure is higher and cell viability is reduced during printing	Effects of laser light on cells are not well studied; high printing costs and complex print control systems
Resolution	Medium	Low	High
Application	Irregular or complex three-dimensional structures	Design of complex structures	Different types of organizational structures
Print speed	Fast	Medium	Slow
Cell viability	Medium	Low	High
Cost	Low	Medium	High
Cell density	Low	High	Medium

**Table 2 bioengineering-10-00299-t002:** Commonly used biological materials [19,20].

Materials	Classification	Composition	Strength	Limitations
Polyethylene glycol (PEG) and PEG copolymer	Synthetic biomaterials	One of the most studied and widely used biomaterials	Can be modified or combined with other biomaterials to design in vitro models Biocompatible, biodegradable, and reproducible Possesses experimental control properties	Failiure to support desired cell behaviors and tissue formation
Collagen	Natural biological materials	Main component of ECM, including Type I and Type IV collagen	Porous structure Strong hydrophilicity	Protein concentration is affected by the biological origin No chemical modifications Biochemical cues cannot be provided
Matrix	Natural biological materials	Tumor-derived product extracted from mouse sarcomas composed of basement membrane components	Provides growth factor information Stimulates cell–matrix interactions Induces differentiation	Lack of biomimetic function in vivo
Complex biomaterials	Complex biomaterials	Consists of natural and synthetic biomaterials	Leads to the formation of advanced in vitro models that resemble in vivo tissues	More expensive Difficult to determine the optimal ratio

**Table 3 bioengineering-10-00299-t003:** Differences between the three types of biological materials [2,11,19].

Characteristic	Natural Biomaterials	Synthetic Biomaterials	Composite Biomaterials
Definition	Composed of naturally occurring substances	Composed of synthetic biological materials	Combination of different kinds of biomaterials
Classification	Protein biomaterials Polysaccharide biomaterials Natural nanomaterials	Polyethylene glycol Polyethylene glycol copolymer	Nanopolymer biomaterials Non-nanopolymer biomaterials
Strength	Better biocompatibility Better biodegradability Provides biological cues Replicates specific ECMs	Easy to control Biocompatible, biodegradable and reproducible A combination of materials is used Cost is relatively low	Better biocompatibility Better thermal stability and antibacterial efficacy Associated applications function more effectively
Limitations	Difficult to control Lacks mechanical integrity Difficult to separate	Toxicity Immune-related issues	More complicated process

## Data Availability

Data sharing is not applicable.

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
