# Peer review of "3D Printing and Its Current Status of Application in Obstetrics and Gynecological Diseases"

_bioengineering, 2023, doi:10.3390/bioengineering10030299_

Round 1

Reviewer 1 Report

In this manuscript, the authors present a review on the use of 3D bioprinting technology in the field of obstetrics and gynecology. This should be an interesting topic on 3D bioprinting technology for biomedical applications. Nevertheless, the whole manuscript is not very well written and must be further improved. Following are my questions and comments to serve as a starting point for reworking this manuscript:

1. In my view, the topic of this manuscript has been well reviewed in previous review papers within the most recent three years. For example, 

A. Daniilidis, Angelos, Theodoros D. Theodoridis, and Grigoris F. Grimbizis. "3D printing in gynecology and obstetrics." 3D Printing: Applications in Medicine and Surgery Volume 2. Elsevier, 2022. 141-157.

B. Kudryavtseva, Elena, et al. "Advantages of 3D printing for gynecology and obstetrics: brief review of applications, technologies, and prospects." 2020 IEEE 10th International Conference Nanomaterials: Applications & Properties (NAP). IEEE, 2020.

C. Yasli, Mert, et al. "Additive manufacturing and three-dimensional printing in obstetrics and 

So please justify what is the advancement or difference of this manuscript?

2. Abstract: Actually, most of the present abstract is introducing 3D printing, and it seems only the last sentence mentions “the use of 3D bioprinting technology in the field of obstetrics and gynecology”. This obviously deviates from the gist of this review.

3. Table 2: The authors only listed commonly used natural biological materials for 3D printing. Nevertheless, synthetic biomaterials and complex biomaterials, rather than natural biological materials, are the most commonly used 3D printing materials for biomedical applications. Please add these materials to the table as well.

4. Only one reference (Ref 19) in Table 3? More relevant original research publications should be added to Table 3.

5. 2. Applications of 3D printing technology in obstetrics and gynecology: I recommend adding a table to summarize the applications of 3D printing technology in obstetrics and gynecology with relevant references added.

6. 2. Applications of 3D printing technology in obstetrics and gynecology: As the main part of this review, this section looks like a literature report rather than a review paper. The authors just briefly summarized some publications regarding 3D printing technology for the treatment of obstetrics and gynecological diseases. By doing so, the readers cannot have a deep impression of the scientific progress in this field. It should be noted that a review paper does not simply summarize the literature you reviewed. You should explain how those results and progress shape our current understanding of the topic and present a clear statement of what you intend to prove by this review. I suggest this section should be seriously polished.

7. I noticed that only two Figures were presented in the Conclusions but none in the main section. I recommend more experimental figures regarding the topic of “3D printing technology in obstetrics and gynecology” that are reported in the cited references should be added in Section 2.

Author Response

Dear Reviewer:

Thank you for your comments concerning our manuscript entitled“Current status of application of 3D printing in obstetrics and gynecological diseases”(ID: bioengineering-2146047). Those comments are all valuable and very helpful for revising and improving our paper. We have studied comments carefully and have made correction which we hope meet with approval. Revised portion are manked in red in the paper. The main corrections in the paper and the responds to the reviewer’s comments are as flowing:

Point 1: In my view, the topic of this manuscript has been well reviewed in previous review papers within the most recent three years. For example, 

  1. Daniilidis, Angelos, Theodoros D. Theodoridis, and Grigoris F. Grimbizis. "3D printing in gynecology and obstetrics." 3D Printing: Applications in Medicine and Surgery Volume 2. Elsevier, 2022. 141-157.
  2. Kudryavtseva, Elena, et al. "Advantages of 3D printing for gynecology and obstetrics: brief review of applications, technologies, and prospects." 2020 IEEE 10th International Conference Nanomaterials: Applications & Properties (NAP). IEEE, 2020.
  3. Yasli M, et al. Additive manufacturing and three-dimensional printing in obstetrics and gynecology: a comprehensive review. Arch Gynecol Obstet. 2023 Jan 13. doi: 10.1007/s00404-023-06912-1. Epub ahead of print. PMID: 36635490.

So please justify what is the advancement or difference of this manuscript?

Response 1: Thanks for your suggestion. The advancement or difference of this manuscript are as follows:

On one hand, the structure of this manuscript mainly consists of two parts: 3D printing technology and its application in obstetric and gynecological diseases. In the section of application, we take different diseases as sub-section standards. We describe the specific application and development of 3D printing technology in each sub-section in detail to provide more ideas for the clinical diagnosis and treatment.

On the another hand, the content of this manuscript includes clinical practice, basic experimental(for example, in the section of intrauterine adhesions, malignant tumors, etc., we summarized the effects of different concentrations of relevant biomaterials on cell culture and cell survival)and medical education issues according to the references in the past 5 years in gynecology and obstetrics. Also, we are looking forward to diseases that are not covered by 3D printing technology, such as twin transfusion syndrome can combine 3D priting technology with it to improve to improve clinical outcomes. We hope our manuscript can help to promote the combination of 3D printing technology and obstetrics and gynecology related diagnosis and treatment, so as to serve patients better.

Point 2: Abstract: Actually, most of the present abstract is introducing 3D printing, and it seems only the last sentence mentions “the use of 3D bioprinting technology in the field of obstetrics and gynecology”. This obviously deviates from the gist of this review.

Response 2: Thank you for your advice, this is very important for the revision of the article. We have changed the abstract to“3D printing technology refers to a new method of using computer-generated three dimensional models for drawing, assembling special bioinks, and manufacturing artificial organs and biomedical products. In recent years, it has gradually become a relatively mature disease treatment strategy, and has been widely used in clinical and basic researchs . And in the field of obstetrics and gynecology, 3D printing technology has been applied in many aspects including disease diagnosis, treatment, pathogenesis research, medical education. Especially in the common gynecological and obstetrical diseases such as intrauterine adhesions, uterine tumors, congenital malformations, and fetal congenital abnormalities,the researchers had obtain rich application experience. This review will systematically summarize the application research of 3D bioprinting technology in the field of obstetrics and gynecology. ”(lines 10-20 of the revised manuscript).

Point 3:Table 2: The authors only listed commonly used natural biological materials for 3D printing. Nevertheless, synthetic biomaterials and complex biomaterials, rather than natural biological materials, are the most commonly used 3D printing materials for biomedical applications. Please add these materials to the table as well.

Response 3: Thank you for your suggestions. We have added the relevant biological materials and related content corresponding to synthetic biomaterials and complex biomaterials to Table 2. We hope that it is clear.

Point 4: Only one reference (Ref 19) in Table 3? More relevant original research publications should be added to Table 3.

Response 4: Thank you for the suggestion, by re-reading the relevant literature, the relevant references in Table 3 also have Ref 2 and Ref 11.

Point 5:Applications of 3D printing technology in obstetrics and gynecology: I recommend adding a table to summarize the applications of 3D printing technology in obstetrics and gynecology with relevant references added.

Response 5: Thank you for your suggestions for the modification of our manuscript, we have added “Table 4 The applications of 3D printing technology in obstetrics and gynecology” to summarize the application of 3D printing technology in different obstetrics and gynecology diseases.  

Point 6: Applications of 3D printing technology in obstetrics and gynecology: As the main part of this review, this section looks like a literature report rather than a review paper. The authors just briefly summarized some publications regarding 3D printing technology for the treatment of obstetrics and gynecological diseases. By doing so, the readers cannot have a deep impression of the scientific progress in this field. It should be noted that a review paper does not simply summarize the literature you reviewed. You should explain how those results and progress shape our current understanding of the topic and present a clear statement of what you intend to prove by this review. I suggest this section should be seriously polished.

Response 6: Thank you for your comments, it is very important to improve the level of the article and the reader's understanding of this topic. By reading and summarizing the references again, we have revised the text to address your concerns and hope that it is now clearer(Article revision mode).

Point 7: I noticed that only two Figures were presented in the Conclusions but none in the main section. I recommend more experimental figures regarding the topic of “3D printing technology in obstetrics and gynecology” that are reported in the cited references should be added in Section 2.

Response 7: Thanks for the suggestion. For various reasons, the author's authorization cannot be obtained. So more experimental figures cannot be provided. Instead, we added relevant experimental or clinical data to enrich the article.

Reviewer 2 Report

Dear Authors

Dear Editor,

The manuscript is of real interest for the auditorium, the  section 2. Applications of 3D printed technology technology in obstetrics and gynecology being very interested but, before publishing, several issues have to be additionally checked. 

1. Table 1. The resolution is high for all which is misleading. I am sure you can find some characteristics. 

2. In table 2 the MATRIX is not at al clear! What does it represent - it have to be clear also for material scientists like me.

3. Table 3 is misleading. Composite materials can be also Natural or synthetic so, it is not clear for me why you added them separately.

4. Figure 3 is just partially visible. there are a lot or sections marked with red so I am asking if this is a draft or it is a resubmission/revised form. 

Minor comments: The english is reasonable but an additional check would be useful, especially please avoid the use of personal (help us, ...)

Best regards,

R1

Author Response

Dear Reviewer:

Thank you for your comments concerning our manuscript entitled“Current status of application of 3D printing in obstetrics and gynecological diseases”(ID: bioengineering-2146047). Those comments are all valuable and very helpful for revising and improving our paper. We have studied comments carefully and have made correction which we hope meet with approval. Revised portion are manked in red in the paper. The main corrections in the paper and the responds to the reviewer’s comments are as flowing:

Point 1: Table 1. The resolution is high for all which is misleading. I am sure you can find some characteristics. 

Response 1: Thanks for the suggestion. By reading through the references, we found that Inkjet bioprinting, Extrusion bioprinting and Laser-assisted bioprinting’s resolution is medium, low and high. However, the resolution is affected by the printing material, the operating temperature, and the size of the nozzle.

Point 2:  In table 2 the MATRIX is not at al clear! What does it represent - it have to be clear also for material scientists like me.

Response 2:Thank you for your advice, it is important for everyone to understand the content of the article. MATRIX represents “A tumor-derived product extracted from mouse sarcomas comprised of basement membrane components ”,We have made changes in Table 2. 

Point 3: Table 3 is misleading. Composite materials can be also Natural or synthetic so, it is not clear for me why you added them separately.

Response 3: Thanks for your suggestion.“Composite materials”in Table 3 refers to "Combination of different kinds of biomaterials", which include the combinations of natural biomaterials, combinations of synthetic materials, and mixtures of natural and synthetic biomaterials. There biological properties differ from each other, so we add them separately.

Point 4: Figure 3 is just partially visible. there are a lot or sections marked with red so I am asking if this is a draft or it is a resubmission/revised form. 

Response 4: Thanks for your suggestions, we've updated and adjusted the image position.

Point 5: Minor comments: The english is reasonable but an additional check would be useful, especially please avoid the use of personal (help us, ...)

Response 5: Thank you for your advice, we have invited native English speakers to help us with manuscript revisions.We have revised the text to address your concerns.The certificate is in the document.

Reviewer 3 Report

Review comments are attached

Author Response

Dear Reviewer:

Thank you for your comments concerning our manuscript entitled“Current status of application of 3D printing in obstetrics and gynecological diseases”(ID: bioengineering-2146047). Those comments are all valuable and very helpful for revising and improving our paper. We have studied comments carefully and have made correction which we hope meet with approval. Revised portion are manked in red in the paper. The main corrections in the paper and the responds to the reviewer’s comments are as flowing:

Point 1: Based on the title of the review article, one would expect that 3D printing and its applications in obstetrics and gynecological pathologies would be the subject of the introduction. Instead, what one sees is that the terms obstetrics and gynecology were mentioned only in the last statement of section 1.3 of the Introduction. The authors correctly introduced 3D bioprinting in the menu of the article yet they need to do that in the context of what has been heralded in the title of the manuscript.

Response 1: Thank you for your suggestions for article modifications, we have changed the title“Current status of application of 3D printing in obstetrics and gynecological diseases”to“3D printing and its current applications in obstetrics and gynecology”.

Point 2: The statement in the introduction of the manuscript, i.e. “Currently, 3D printing technology, developed in the 80s, is a commonly used bioprinting technology, and refers to the basis of specific digital designs created based on data derived from conventional two-dimensional i CT, ultrasound, and MRI images, after processing and format conversion, through several steps, to obtain solid 3D models.” should be corrected to read “Currently, 3D printing technology, developed in the 80s, is a commonly used bioprinting technology, and refers to the basis of specific digital designs created through data derived from conventional two-dimensional CT, ultrasound, and MRI images, following processing and format conversion, through several steps, to obtain solid 3D models.”.

Response 2: Thanks for your advice. Since the acronyms CT, MRI ,US did not appear before, so we spell them out here,and change the above expression to“Currently, 3D printing technology, developed in the 80s, is a commonly used bioprinting technology and refers to the basis of specific digital designs created through data derived from conventional two-dimensional Computed Tomography (CT), Ultrasound (US), and Magnetic resonance imaging (MRI) images, following processing and format conversion, through several steps, to obtain solid 3D models.”( lines 36–41 of the revised manuscript).

Point 3: In Table 2, the head table reads “Constitute”. What do the authors mean by it? Do they mean “Composition”? Corrections should apply.

Response 3: Thanks for your advice. We have changed "Constitute" to "Composition".

Point 4: In Table 2, the last entry on the Table head is listed as “limitations”. It should be “Limitations”, in line with the remainder of the head entries.

Response 4: Thanks for your advice. We have changed "limitations" to "Limitations".

Point 5: In Table 2, on the left hand side column, the bottom item is listed as “Combination of collagen and the matrixed”. It is not clear what the authors mean by “matrixed”. Matrixed what? Ample clarification and corrections should apply.

Response 5: Thanks for your suggestion, we found the interpretation of “composite biomaterials” to be relatively inaccurate. Therefore, we have made some adjustments to Table 2. We changed "Combination of collagen and the matrixed" to "Consist of natural and synthetic biomaterials" and made the corresponding changes in the table. We hope that it is now clearer.

Point 6: In Table 3, in the Natural Biomaterials column describing Strength, the description “It provide biological cues, and the replication of specific ECM components is possible” should be corrected to read “It provides biological cues, and replication of specific ECM components is possible”.

Response 6: Thanks for your suggestion. In order to keep with the above format, we finally decided to change it to "Provide biological cues, Replicate specific ECM". 

Point 7:In section 2.1.1 “Uterine fibroids”, the acronymic terms (US, CT, MRI,) listed in statement “In combination with preoperative US, CT, MRI, and other related examinations,” should be spelled out.

Response 7: Thank you for your suggestions for the revision of the article, since the acronyms CT, MRI, US have appeared in the revised introduction section( lines 39–40 of the revised manuscript), so this article has not been modified here.

Point 8: In the section on endometrial cancer, the statement “This model would improve the effectiveness and accuracy of preoperative planning and patients' understanding of their disease, surgical process, and risk of complications. The postoperative recurrence of endometrial cancer occurs most commonly in the vaginal area, and personalized 3D molds can help us to provide personalized treatment for individuals undergoing postoperative chemotherapy” should be corrected to read “This model would improve the effectiveness and accuracy of preoperative planning and patient understanding of their disease, surgical process, and risk of complications. Postoperative recurrence of endometrial cancer occurs most commonly in the vaginal area, and personalized 3D molds could help provide personalized treatment to individuals undergoing postoperative chemotherapy”.

Response 8: Thank you for your opinion, we have changed the above expression to“This model would improve the effectiveness and accuracy of preoperative planning and patients' understanding of their disease, surgical process, and risk of complications. Postoperative recurrence of endometrial cancer occurs most commonly in the vagnal area, and personalized 3D molds could help provide personalized treatment to individuals undergoing postoperative chemotherapy ”(lines 350–355 of the revised manuscript).

Point 9: In section 2.2 premature ovarian failure, the statement “Joo et al. [36] created collagenrich biomimetic 3D shells and cultured rodent ovarian follicles in various concentrations of type I collagen hydrogels from 1% to 7%, it comes that in a relatively high concentration of collagen produces denser matrix compared to lower concentrations of collagen.” is too long and not understood. It should be split into shorter statements that are comprehensible.

Response 9: Thank you for your opinion, we have changed the expression to“Joo et al. [36] created collagen-rich biomimetic 3D shells and cultured rodent ovarian follicles in various concentrations of type I collagen hydrogels. And the concentration of the collagen hydrogels is from 1% to 7%. The outcome comes that in a relatively higher concentration of collagen produces denser matrix.”(lines 447–450 of the revised manuscript).

Point 10: In section “2.3 Intrauterine adhesions”, the statement “IUA formation is directly related to damage to the basal layer, where endometrial stem cells are located Hysteroscopic intrauterine adhesiosiotomy (TCRA) is commonly performed in individuals with IUAs, but re-adhesion may occur after surgery.” is not understood and appears that something is missing (a period somewhere in order to make sense?). The statement should be split into shorter comprehensible statements.

Response 10: Thank you for your advice, we have changed the expression to“The formation of intrauterine adhesions is directly related to the damage of the basal layer,where endometrial stem cells are located. Hysteroscopic intrauterine adhesiosiotomy (TCRA) is the main method for the treatment of IUA. However, re-adhesion may occur after TCRA”(lines 475–478 of the revised manuscript). We hope that it is now clearer and easier to understand.

Point 11:In the ensuing paragraph of the same section, the statement “With regarding to IUA treatment, current studies have proven that granulate colony stimulating factor (G-CSF) has a positive effect” should be corrected to read “With regard to IUA treatment, current studies have proven that granulocyte colony stimulating factor (G-CSF) has a positive effect”.

Response 11: Thank you for your advice, we have changed the expression to“With regard to IUA treatment, current studies have proven that granulate colony-stim ulating factor (G-CSF) has a positive effect”(lines 526–527 of the revised manuscript). 

Point 12: Figure 3 cannot be seen and looked at as a whole. It should be resized so as to be visualized.

Response 12: Thanks for your suggestions, we've updated and adjusted the image position.

Round 2

Reviewer 1 Report

The authors have addressed my concerns and improved this manuscript accordingly. I have no more comments.

Author Response

Dear Reviewer:

Thank you for your comments concerning our manuscript entitled“3D printing and its current status of application in obstetrics and gynecological diseases”(ID: bioengineering-2146047). We have invited native English speakers to help us with manuscript revisions for the second time.We have revised the text to address your concerns.

Reviewer 2 Report

Dear Authors,

Dear All,

During the revision in table 2 you wrote Olyethylene glycol (PEG) instead of Polyethylene glycol (PEG).

Best regards,

R1

Author Response

Dear Reviewer:

Thank you for your comments concerning our manuscript entitled“3D printing and its current status of application in obstetrics and gynecological diseases”(ID: bioengineering-2146047). Your comment is valuable and very helpful for revising and improving our paper. We have studied comment carefully and have made correction which we hope meet with approval. Revised portion are manked in red in the paper. The main corrections in the paper and the responds to the reviewer’s comments are as flowing:

Point:During the revision in table 2 you wrote Olyethylene glycol (PEG) instead of Polyethylene glycol (PEG).

Response : Thanks for your suggestion. We have changed "Olyethylene glycol" to "Polyethylene glycol".

As for the English language, we have invited native English speakers to help us with manuscript revisions for the second time.We have revised the text to address your concerns.

Reviewer 3 Report

Review comments are attached

Author Response

Dear Reviewer:

Thank you for your comments concerning our manuscript entitled“3D printing and its current status of application in obstetrics and gynecological diseases”(ID: bioengineering-2146047). Those comments are all valuable and very helpful for revising and improving our paper. We have studied comments carefully and have made correction which we hope meet with approval. Revised portion are manked in red in the paper. The main corrections in the paper and the responds to the reviewer’s comments are as flowing:

Point 1: In the introduction, the statement “It is an interdisciplinary science that is closely related to the fields such ad medicine, biology, …” should be corrected to read “It is an interdisciplinary science closely related to the fields of medicine, biology, …”.

Response 1:Thank you for your suggestion, we have changed the above expression to“This interdisciplinary science closely related to the fields of medicine, biology, …”(lines 26-27 of the revised manuscript).

Point 2: The legend of Table 1 “Table 1. Differences in various of bioprinting process [10-12]” should be corrected to read “Table 1. Differences in various types of bioprinting processes [10-12]”.

Response 2:Thank you for your suggestion, we have changed the legend of Table 1 to“Table 1. Differences in various types of bioprinting processes [10-12]”(line 110 of the revised manuscript).

Point 3: In section “2.1.2. Malignancies” the statement “Malignant tumors of the female reproductive system are an important cause of female. death, and the mechanism associated with their occurrence is still unclear.” should be corrected to read “Malignant tumors of the female reproductive system are an important cause of female death, with the mechanism associated with their occurrence being still unclear.”.

Response 3:Thank you for your opinion, we have changed the above expression to“ Malignant tumors of the female reproductive system are an important cause of female mortality, with the mechanism associated with their occurrence being still unclear”(lines 215-216 of the revised manuscript).

Point 4: In the section on endometrial cancer, the herein amended statement “This model would improve the effectiveness and accuracy of preoperative planning and patients' understanding of their disease, surgical process, and risk of complications.” should again be corrected to read “This model would improve the effectiveness and accuracy of preoperative planning and understanding on behalf of the patient of their disease, surgical process, and risk of complications.”. 

Response 4:Thank you for your advice, we have changed the expression to“This model would improve the effectiveness and accuracy of preoperative planning and understanding on behalf of the patient of their disease, surgical process, and risk of complications.”(lines 185-187 of the revised manuscript).

Point 5: In section 2.2 premature ovarian failure, the suggested change is not there in the manuscript. To that end, it is again suggested that the current state of the statement “Different concentrations of biological material can also affect cell growth. Joo et al. [36] created a bionic 3D shell rich in collagen for culturing rodent ovarian follicles in various concentrations of type I collagen from 1% to 7%. The outcome comes that in a relatively high concentration of collagen produces denser matrix.” be changed to “Different concentrations of biological material can also affect cell growth. Joo et al. [36] created a bionic 3D shell, rich in collagen, for culturing rodent ovarian follicles in various concentrations of type I collagen from 1% to 7%. The outcome shows that in a relatively high concentration of collagen a denser matrix is produced”.

Response 5:Thank you for your advice, we have changed the expression to“The cell growth in 3D bioprinting is influenced by the concentration of biological material. Joo et al. [36] created a bionic 3D shell,rich in collagen,for culturing rodent ovarian follicles in various concentrations of type I collagen ranging from 1% to 7%. The study showed that in a relatively high concentration of collagen a denser matrix is produced ”(lines 840-843 of the revised manuscript). 

As for the English language, we have invited native English speakers to help us with manuscript revisions for the second time.We have revised the text to address your concerns.